# Factors Affecting Roadkills of Semi-Aquatic Mammals

**DOI:** 10.3390/biology11050748

**Published:** 2022-05-13

**Authors:** Linas Balčiauskas, Jos Stratford, Andrius Kučas, Laima Balčiauskienė

**Affiliations:** Nature Research Centre, Akademijos 2, 08412 Vilnius, Lithuania; birding.jos@gmail.com (J.S.); andrius.kucas@gamtc.lt (A.K.); laima.balciauskiene@gamtc.lt (L.B.)

**Keywords:** European beaver, Eurasian otter, American mink, muskrat, roadkill, Lithuania

## Abstract

**Simple Summary:**

Semiaquatic mammal roadkill, specifically the European beaver, Eurasian otter, American mink and muskrat was analyzed in Lithuania in the period 2002–2021. From almost 40,000 recorded roadkill cases for all species, these species were registered 60, 22, 26 and 3 times, respectively, thus being rare events. Registered beaver roadkill numbers correlated with an increased registration effort and traffic intensity, while otter roadkill correlated with a registration effort only, and mink roadkill correlated with an increased population (estimated through its proxy and the number of hunted individuals per year). The roadkill was not always in close proximity to waterbodies, with 38–54% of roadkill of these species occurring over 200 m from the nearest water source. With the American mink and muskrat being invasive species in the EU and the Eurasian otter protected in many countries, it could be considered valuable to enhance the registrations of their roadkill (using targeted efforts by drivers, hunters or other citizen scientists) in order to obtain the extrapolated amount of roadkill and to use this knowledge in species management.

**Abstract:**

We previously showed that registration efforts and traffic intensity explain 90% of variation in mammal roadkill numbers, 70% of variation in the numbers of recorded species and 40% of diversity variation. Here we analyze semiaquatic mammal roadkill in Lithuania in 2002–2021, relating these to the monitoring effort. From 39,936 analyzed roadkill, the European beaver (*Castor fiber*) was registered 60 times, American mink (*Neovison vison*) 26 times, otter (*Lutra lutra*) 22 times and muskrat (*Ondatra zibethica*) 3 times. The average roadkill indexes were 0.000065, 0.00076, 0.00061 and 0.00010 ind./km/day, and the extrapolated annual roadkill for the country was 44–357, 36–456, 49–464 and 89–144 individuals, respectively. Beaver roadkill numbers correlated with the registration effort and traffic intensity, otter roadkill with registration effort only and mink with hunting bag (number of hunted individuals per year). Roadkill was not always related to proximity to water, with 38–54% of roadkill occurring over 200 m from the nearest water source. With American mink and muskrat being invasive species in the EU and otter protected in many countries, it is valuable to enhance the registrations of their roadkill (using targeted efforts by drivers, hunters or other citizen scientists) to obtain the extrapolated amount of roadkill and to use this knowledge in species management.

## 1. Introduction

Semiaquatic mammals form an ecological group of animals whose lifestyles are related both to aquatic and terrestrial habitats with varying degrees of dependence [1,2]. Four such medium-sized species are present in Lithuania, namely European beaver (*Castor fiber*), Eurasian otter (*Lutra lutra*), American mink (*Neovison vison*) and muskrat (*Ondatra zibethicus*) [3], the list of species being similar to that in North America [4]. Of these species, *C. fiber* is considered an ecosystem engineer as it modifies ecosystems and enhances habitat suitability for the other animals [5,6].

The habitats used by this group of mammals include various waterbodies, such as swamps, streams, rivers and lakes [7] with seasonal water level changes having a significant influence [4]. Urban and peri-urban landscapes are also being inhabited [8]. In Lithuania, *C. fiber* is widespread and inhabits nearly all waterbodies across the country, especially those with shores overgrown with deciduous trees and shrubs. It is abundant in land reclamation channels and builds dams to raise water levels where required. Even small pools and swamps in agricultural landscapes are inhabited, at least temporarily, while there are food sources available [9]. *O. zibethicus* inhabits waterbodies abounding in water vegetation, such as swamps, lakes and slow river ox-bows. *L. lutra* is also widespread and inhabits, at least temporarily, all types of waterbodies with running water, including reclamation channels, as well as lakes and ponds. *N. vison* is found in river valleys and inhabits ponds and swampy areas [10,11]. The latter two species are characterized by high habitat plasticity and by their frequent use of beaver-made dams, lodges and burrows [12]. Habitat use by all of these species is not stable but changes over the medium-term period, e.g., over decades in relation to population changes [13].

Among these semiaquatic mammals, *C. fiber* is a game species with an unlimited quota of hunted animals per year, hereafter referred to as “hunting bag”. Hunting totals amounted to 19,997, 19,907 and 15,561 individuals in 2019, 2020 and 2021. *N. vison* and *O. zibethicus* are invasive species in Lithuania, and there are no limits on hunting, though the hunting bag is quite small. In 2019–2021, the annual hunting bag of *N. vison* was 42–53, while that of *O. zibethicus* decreased from 105 ind. in 2019 to 23 ind. in 2021. *L. lutra* is no longer listed in the *Red Data Book* of Lithuania [14], but it is not hunted.

In Lithuania and neighboring Poland, semiaquatic mammals are not frequently involved in wildlife–vehicle collisions (WVCs) [15,16]. Despite different species compositions, this is also true in countries with quite different climates [17,18,19,20,21]. A notable exception, however, is coypu (*Myocastor coypus*), the most frequent (67.6% of casualties) roadkill mammal in the central River Po plain in Italy [22].

Despite the scarcity of semiaquatic mammals in WVCs, they are, however, the highest single cause of mortality for *L. lutra* in most European countries [23]. The amount of general roadkill has shown a tendency to increase in recent years, both in Lithuania [24] and in other European countries [15,25,26,27] and elsewhere ([28,29,30], etc.).

Several groups of factors are involved in WVCs, these concentrating in some locations to form WVC hotspots [15,30]. The first group of factors relates to transport and driver variables, these including the annual average daily traffic volume (AADT) and speed [16,31], vehicle types [16,32] and driver behavior [33]. The second group of factors are those pertaining to the mammal species themselves, most particularly the population densities of the roadkill species [34]. The two above-mentioned factors, specifically traffic intensity and population size, explained 63–93% of the variation in roadkill numbers in ungulates in Lithuania [16]. The third group of factors includes landscape and terrain factors, such as habitat composition and distances to certain habitats ([35,36,37,38] and references therein). The fourth group includes time factors, including season and time of day [25,39,40], and weather factors [29].

Not all investigations have confirmed the relationship between roadkill frequencies and population sizes, for example [15]. This cited investigation, however, was based on a small-scale analysis and used population numbers only in the nearest hunting districts. Similarly, a low correlation was also found on a local scale in Lithuania between raccoon dog (*Nyctereutes procyonoides*) numbers and roadkill [38]. Moreover, habitat preferences in semiaquatic mammals may have some specificity: the best survival might be observed not in the most frequently selected habitats. In addition, changes in territory use in these species might be rapid and significant [13].

The aim of this paper is to present long-term data on semiaquatic mammal roadkill in Lithuania, including extrapolating to estimate national roadkill numbers. For this study, we looked at the four affected species, namely *C. fiber*, *L. lutra*, *N. vison* and *O. zibethicus.* We tested (i) if semiaquatic mammal roadkill was related to proximity to waterbodies (rivers and lakes) or, in the cases of *C. fiber* and *O. zibethicus*, to wetlands and (ii) if there is any relationship among roadkill frequencies and population/hunting bag sizes, transport intensity and sampling effort.

## 2. Material and Methods

We used data collected by professionals at the Nature Research Centre. A registration session was defined as registering roadkill along any length of a same-numbered road travelled on the same day. A total of 3570 such sessions were conducted nationally from 2007–2021, with the total length of all driven routes equaling 275,300 km. The most frequent registration sessions were performed on the main roads A1, A2, A14 and A8 (405, 405, 400 and 210 sessions, respectively). The number of sessions on other main, national and regional roads (Figure 1) varied from 1–125. Data collected by the Lithuanian Police Traffic Supervision Service between 2002 and 2021 were also used. In total, we analyzed data on 39,936 WVCs involving wild mammals (MVCs). In both datasets, individuals not identifying with the species level (including “mustelid”, “small carnivore”, “hare”, “big game”, etc.) were pooled under category “unknown”.

Domestic mammal roadkill (*n* = 3150) was not analyzed in detail, as these species are mostly related to settlements not habitats. The most frequent roadkill in this category was dogs (*n* = 2134), cattle (*n* = 414), horses (*n* = 220) and cats (*n* = 220), followed by sheep, goats, pigs and rabbits in descending numbers.

In 2021, there were 21,237 km of state roads of national significance, these being 1751 km of main roads (AADT 3000–20,000), 4928 km of national roads (AADT 500–3000) and 14,559 km of regional roads (AADT up to 500) [41,42]. Changes in road lengths since 2017 [43] have been negligible and, thus, were ignored.

Roadkill indices were calculated as the number of killed animals per km/day. “Empty routes”, where roadkill had already possibly been removed, were included into the yearly index. Estimations of the number of animals killed at the country level for each of the four species were based on the length of the season of actual registrations (Figure 2). *L. lutra* and *N. vison* were not registered in January and February; therefore, the season length was set at 300 days, while *C. fiber* was not registered in February, and thus the season length was set at 330 days. Based on the ecology of species [44] and the registered roadkill, the season length for *O. zibethicus* was set at 240 days. Calculations were done using the total lengths of the main and national roads only (6679 km) as the roadkill registration efforts on the regional roads were not sufficient. The presented estimations can, therefore, be seen as minimum numbers.

Data on population size and hunted numbers of *C. fiber* and hunted numbers of *N. vison* and *O. zibethicus* are presented in Table 1. This information was collected over a longer time period, but earlier information is not available as open data anymore [45]. *N. vison* and *O. zibethicus* are not included in game survey programs; therefore, only information on the bag size was available. *L. lutra* is a protected species; therefore, there is no hunting bag, and the species is not included in Table 1. As a result, the numbers of *L. lutra* and *N. vison* that we used are based on monitoring data and the distribution of these species along rivers [12,46,47,48].

In summary, the population of *C. fiber* in the years since 2010 has ranged from 40 to 50 thousand individuals (Table 1), while that of *L. lutra* has been 3000–5000 individuals and that of *N. vison* in the region of 5000–10,000 individuals [48,49]. The numbers of *O. zibethicus* have sharply declined since the 1990s and based on the reduced hunting bag, could be 2000–3000 in the last decade [48].

Data on annual average daily traffic (AADT) for the roads with semiaquatic mammal roadkill were obtained from the Lithuanian Road Administration under the Ministry of Transport and Communications [42].

Distances from the roadkill to the nearest waterbody were investigated at the (1) medium spatial scale using the Corine land cover inventory (CLC) 1:50,000 [50] and (2) fine scale using the spatial dataset of (geo) reference base cadastre (GRPK) 1:10,000 (https://www.geoportal.lt/metadata-catalog/catalog/search/resource/details.page?uuid=%7B513C0C29-0447-CB3D-4585-2390144D20D2%7D accessed on 10 February 2022). In the CLC, two land cover classes (411 inland marshes and 412 peat bogs) were used as “wetlands” and two classes of waterbodies (511 water courses and 512 waterbodies) as “water”. Due to the absence of roadkill in the relevant areas, two classes (521 coastal lagoons and 523 sea and ocean) were not analyzed. In the GPRK, we used two habitats, namely “rivers” and “lakes”. Distances from the roadkill points to the closest polygon were measured using near distance measurement tools (https://pro.arcgis.com/en/pro-app/2.8/tool-reference/analysis/near.htm accessed on 5 February 2022) that are part of the standard ArcGIS Proximity toolset (https://pro.arcgis.com/en/pro-app/2.8/tool-reference/analysis/an-overview-of-the-proximity-toolset.htm accessed on 5 February 2022). As distance statistics, species-based averages and SE were calculated. After finding that the distribution of distances did not conform to normal, between-species differences were evaluated with the nonparametric Kruskal–Wallis H test.

At the country scale, we applied a multiple regression analysis to the annual *C. fiber* and *N. vison* roadkill to test the possible cumulative influence of the categorical predictors, specifically annual population and hunting bag sizes, sampling effort and AADT. The confidence level was set as *p* < 0.05. At the local scale, we tested if there was a correlation among roadkill, sampling effort and AADT using the road as the sampling unit. The Pearson correlation coefficient was used. Calculations were done in Statistica for Windows, ver. 6.0 (StatSoft, Inc., Tulsa, OK, USA).

The proportion of semiaquatic mammal roadkill and the 95% CI for species proportion among all wild mammal roadkill were calculated with the Wilson method of the score interval [51] using OpenEpi epidemiological software [52]. Differences in the proportions of roadkill were evaluated using the G test online calculator [53].

## 3. Results

### 3.1. Species Composition of the Wild Mammal Roadkills

In the analyzed roadkill, 32 wild mammal species were registered in 2002–2021 (Table 2). Unidentified species accounted for 8.30%, with the pooled number for roadkill being < 20 per species – 0.12% (Table 2). Semiaquatic mammals accounted for only a marginal proportion of the total amount of roadkill registered, this being 0.15% of all roadkill (CI = 0.12–0.19%) for *C. fiber*, 0.065% (CI = 0.044–0.095%) for *N. vison*, 0.055% (CI = 0.036–0.083%) for *L. lutra* and 0.0075% for *O. zibethicus*. Based on the G-test (G = 62.43), the proportion of *C. fiber* roadkills was higher (*p* < 0.001) and that of *O. zibethicus* lower (*p* < 0.003) than the other two species.

### 3.2. Spatiotemporal Distribution and Index of Semiaquatic Mammal Roadkill

The spatial distributions of the semiaquatic roadkill species show *C. fiber* registrations to be scattered across the county (Figure 3a), while *L. lutra* roadkill was largely absent in the north of the country (Figure 3b). Most *N. vison* roadkill was concentrated in the eastern part of the country (Figure 3c), especially on the main road A14 (Figure 1), this characterized by the highest sampling effort.

There were no differences in the annual proportions of the *N. vison* (G = 19.5, NS), *O. zibethicus* (G = 2.3, NS) or *L. lutra* (G = 11.0, NS). *C. fiber* roadkills has increased in number in the last decade (Figure 4), and annual differences are significant (G = 72.3, *p* < 0.0001), the most notable increase being in 2021 (G = 32.8, *p* < 0.0001).

Based on the long-term observations, the average roadkill indexes of the semiaquatic mammals were 0.000062 ± 0.000023 ind./km/day for *C. fiber*, 0.000077 ± 0.000023 ind./km/day for *L. lutra*, 0.000074 ± 0.000015 ind./km/day for *N. vison* and 0.000010 ± 0.000006 ind./km/day for *O. zibethicus*. On average, the amount of roadkill in these species per year was 124 (95% CI = 33–214), 154 (65–243), 149 (85–213) and 16 (0–34) individuals, respectively. The dynamics of the extrapolated roadkill numbers for the country is shown in Figure 5. The annual roadkill of *C. fiber* was between 44 ind. in 2020 and 357 ind. in 2017, that of *L. lutra* between 36 ind. in 2013 and 456 ind. in 2021 and that of *N. vison* between 49 in 2021 and 464 in 2015. The predicted roadkill of *O. zibethicus* was 89–144 individuals.

Due to inaccuracies arising from the limited amount of survey, we exclude the predicted amount of *L. lutra* roadkill in 2004. The figure calculated was in the order of 6000 individuals per year, a strikingly high figure.

### 3.3. Factors Affecting Semi-Aquatic Mammal Roadkills

At the local scale, using the road as the sampling unit, we analyzed correlations between semiaquatic animal roadkill numbers and the sampling effort (Figure 6) and traffic intensity (Figure 7). The enhanced registration effort yielded higher numbers of registered roadkill of *C. fiber* and *L. lutra* (r = 0.9, *p* < 0.001 and r = 0.69, *p* < 0.02), but not *N. vison* (r = 0.53, NS). Most roadkill was registered on a relatively short segment of the A14 main road (Figure 3) in the period of 2007–2021. This was sampled on a weekly basis with exception of the years 2019 and 2020. All three *O. zibethicus* registrations were on roads with high a sampling effort.

Traffic intensity was correlated with *C. fiber* roadkill numbers (r = 0.68, *p* < 0.001), but did not correlate with *L. lutra* or *N. vison* roadkill numbers (r = 0.34 and r = −0.07, both NS; Figure 7). However, the semiaquatic mammals did tend to be killed on the main roads that were characterized by the highest AADT and traffic speed. Such roads accounted for 100% of *N. vison* and *O. zibethicus* roadkill, 77.3% of *L. lutra* and 46.7% of *C. fiber*. National roads, with medium AADT, accounted for 35.0% of *C. fiber* roadkill and 18.2% of *L. lutra* roadkill, while on regional roads with low AADT, the respective values were just 10.0% and 4.5%.

At the country scale, we used regression analysis to see if the amount of annually registered semiaquatic animal roadkill was related to the annual averages of AADT, the sum of sampling effort, the hunting bag size of *C. fiber* and *N. vison* and the population size of *C. fiber*. The model for *C. fiber* was not significant (F_3,14_ = 1.97, *p* = 0.16) and explained less than 20% of roadkill variation (R^2^ = 0.19. Of the four analyzed factors for this species, only AADT was significant (F = 2.39, *p* < 0.05). Population number, hunting bag size and sampling effort did not correlate with the amount of roadkill (r = 0.23, 0.19 and 0.28 respectively, all NS; Figure 8).

The model for *N. vison* was significant (F_3,15_ = 21.3, *p* < 0.0001) and explained nearly 80% of roadkill variation (R^2^ = 0.77). Both the size of the hunting bag and sampling intensity were significant (F = 5.79 and F = 4.47, both *p* < 0.001), while AADT was not (F = 0.43, NS). Numbers of hunted *N. vison* correlated with the roadkill of the same year strongly and positively (r = 0.75, *p* < 0.001; Figure 8) and so did sampling intensity (r = 0.61, *p* < 0.001). *L. lutra* and *N. vison*, although about 10 and 5 times less numerous in terms of population than *C. fiber*, had relatively high levels of roadkill, these still amounting to about half of the *C. fiber* roadkill numbers. In *O. zibethicus*, the amount of registered roadkill was too small to make any comparisons.

We found the differences in the distances from the roadkill to the nearest water source to not be significant among semiaquatic mammal species, regardless of the scale (size) of the water source: CLC water, Kruskal-Wallis H_3,92_ = 1.37, *p* = 0.71; CLC wetland, H = 2.87, *p* = 0.41; GRPK river, H = 3.22, *p* = 0.36 and GRPK lake, H = 0.94, *p* = 0.81. Measured at the highest possible map scale, 26% of *C. fiber* roadkill occurred less than 50 m from the smallest possible water source, 43% less than 100 m and 62% less than 200 m from water. For *L. lutra*, the respective figures were 24%, 38% and 48%, while for *N. vison* they were 15%, 19% and 46%. In other words, 38–54% of roadkill in these species occurred over 200 m from the nearest water (Figure 9). Differences among species were not significant. Distances to large water sources, including wetlands, were much bigger. The average distances are presented in Table 3. 

## 4. Discussion

Upon completing our analysis of roadkill of the four affected semiaquatic mammal species in Lithuania, we found roadkill to be a rare event. The proportion of *C. fiber* was 0.15% of all roadkill on average, with the proportions of *N. vison* and *L. lutra* being 2–3 times lower. Extrapolated from the roadkill index for the entire network of main and national roads of the country, the number of individuals killed annually is 44–357 for *C. fiber*, 36–456 for *L. lutra*, 49–464 for *N. vison* and 89–144 for *O. zibethicus*.

Otter roadkill is of particular interest as this species was decreasing for a long time in many western countries. In Britain, *L. lutra* roadkill amounted to 673 animals in 1971–1996 [54], these numbers increasing later, with 500 records in the southwest of the country alone between 1984 and 2004 [55]. In a 3000 sq. km area of northeastern Germany, there were 88 instances of *L. lutra* roadkill in 1990–2003 [56], while the total amount of *L. lutra* roadkill in eastern Germany was 746 ind. In the period 1957–1998 [57]. In the Czech Republic, there were 316 instances of *L. lutra* roadkill from 1990 to 2011 [58], while 53 ind. Were recorded in Hungary during 2006–2011 [59]. These relatively low numbers are in accordance with those found in Denmark [60], but 109 instances of otter roadkill in northern Israel during 1965–2004 represented an estimated 5% loss of the population each year [61]. Based on our data, the *L. lutra* roadkill numbers in Lithuania account for between 1% and 10% of the population annually, with data from the last three decades not showing any overall population decrease [12,46,48].

Higher traffic intensity is mentioned as a factor for increased *L. lutra* roadkill frequency [58], though our study did not confirm this. At present, neither spatial nor temporal correlations allow us to determine whether or not the number of otter casualties has had an impact on otter populations [55].

In general, the global trend of an increase in roadkill numbers over time is accepted, see references in [15,62], though during the COVID-19 lockdowns, there were decreases in roadkill, at least in some species [63]. In this context, the numbers of semiaquatic animal roadkill in various countries are very low. For various mammal species, low roadkill rates may be explained as due to the species being a habitat specialist [64]. In terms of semiaquatic species, it may be that aquatic and wetland habitats are not frequently around roads.

In Poland, a neighboring country of Lithuania, the proportion of *O. zibethicus* was 0.50% of total mammal roadkill in 2000–2016, while that of *C. fiber* and of *N. vison* was both 0.38% [65]. In the southwestern part of Poland in the period 2001–2003, no *C. fiber* roadkill was registered, while that of *O. zibethicus* was 0.52% and *N. vison* 0.26% out of 383 wild mammals [66]. In northeastern Poland, on the road crossing the valley of the large river Biebrza, *O. zibethicus* and *N. vison* roadkill each accounted for 1.25% out of 80 wild mammal registrations [67].

No semiaquatic mammal roadkill was registered in Croatia between 2005 and 2008 [68] or between 2007 and 2009 [69]. The same absence of semiaquatic mammal roadkill was characteristic to a study of 1175 instances of roadkill in Brittany, western France [70], 40,000 instance of roadkill in Czech Republic [27] and ca. 2500 roadkill registrations in Walonia, Belgium [71].

However, there were also cases with much higher proportions of semiaquatic animal roadkill. In the Republic of Ireland in 2008–2010, out of 548 roadkill instances, 15 instances were *N. vison* and 10 *L. lutra*, comprising 2.74% and 1.82%, respectively [72]. In areas where roads run in river valleys and wetlands, the proportion of *O. zibethicus* roadkill may reach much higher values. There are two investigations from North America that report very high proportions of *O. zibethicus* roadkill—10.4% in NY State [73] and 15.75% in California—in an area where a highway traverses a tidal marsh and diked baylands [74].

Our data show that 38–54% of semiaquatic mammal roadkill occurred more than 200 m from the nearest water. In this, we included the smallest possible water sources, such as land reclamation ditches, which are nearly or completely dry in the warm season. For *L. lutra*, such a phenomenon was known before. In Germany, between 1985 and 1993, 46.84% of otter roadkill was registered at sites with no adjacent waterbodies [75], and 32% were further than 200 m [56]. By contrast, 67% of all *L. lutra* casualty records occurred in a 100 m wide zone surrounding fresh water and the coast in Great Britain [54] and 57% in Israel [61]. Data about the other species have not been published before.

Of the other peculiarities of semiaquatic animal roadkill, it should be mentioned that these events are not representative of the population [76]; there are behavior differences in relation to oncoming vehicles that may lead to differences in their roadkill [77], and possible mitigations can create barrier effects for them [54,73]. It was shown that the survival of the American beaver (*Castor canadensis*) is inverse to habitat suitability; thus, a new approach might be required to understand semiaquatic mammal distribution and movements [78].

In Europe, *N. vison* densities vary from below one to over ten individuals per 10 km of water length [79]. Being a semiaquatic, generalist predator, *N. vison* can live in various habitats: in Greece, two thirds of records were near water, but many animals were also recorded far from water [80]. In central Italy, roadkill was the first evidence of species spread [81]. Avoiding urbanized areas and isolated ponds in the agricultural landscape is characteristic to the species [82]. During the dry season and droughts, *N. vison* readily uses terrestrial habitats, though the species is subjected to increased mortality when away from streams [83]. Under conditions of climate change, the drying of landscapes will affect local hydrology regimes, changing semiaquatic mammal population dynamics and movements [84]; therefore, the amount of roadkill registered not in proximity to waterbodies may further increase. This may apply to *O. zibethicus* as it is the semiaquatic mammal species most restricted to aquatic habitats [83]. Finally, habitat features may be not the main factors affecting occupancy by semiaquatic mammals, at least for the river otter (*Lontra canadensis*) and *N. vison* [85].

The fate of introduced and reintroduced semiaquatic species in Lithuania varied. Following the release from captivity in 1950 and the introduction in 1953, *N. vison* numbers continuously increased [10], totally suppressing the native European mink (*Mustela lutreola*) [11,14].

After the re-establishment and reintroduction of *C. fiber* to Lithuania in the 1940s and 1950s, they rapidly colonized the entire country [9,10]. According to some expert opinions, their numbers have been close to 100,000 individuals in recent decades [10], i.e., twice the official estimate (see Table 2). With the collapse of the fur market, these numbers are near the carrying capacity of the habitats.

*O. zibethicus* was introduced into eastern and southern parts of Lithuania in 1954, where it still remains most abundant. It is scarce in the north of the country [10,11]. Across the country, however, the species declined by about 94% in the period from 1989 to 2013 [86,87]. The success of the initial introduction is considered to be related to similarities to the climate and habitats within the native species range, as well as to the presence of a free ecological niche [86]. By contrast, the reasons for the decline are not well understood, and the presence of a high number of native predators may be involved, as may pathogens [87].

As *N. vison* and *O. zibethicus* are considered invasive species in the EU, while *L. lutra* is protected in many countries of Europe, it could be considered valuable to enhance registrations of their roadkill (using targeted efforts by drivers, hunters or other citizen scientists) in order to obtain the extrapolated amount of roadkill and to use this knowledge in species management.

## 5. Conclusions

Semiaquatic mammal roadkill is rare, even in *C. fiber* accounting for only 0.15% of total roadkill numbers. The national annual roadkill numbers were 44–357 in the case of *C. fiber*, 36–456 in *L. lutra*, 49–464 in *N. vison* and 89–144 in *O. zibethicus*. Roadkill does not necessarily occur in proximity to waterbodies or wetlands, with 38–54% occurring over 200 m from the nearest water sources.

## Figures and Tables

**Figure 1 biology-11-00748-f001:**
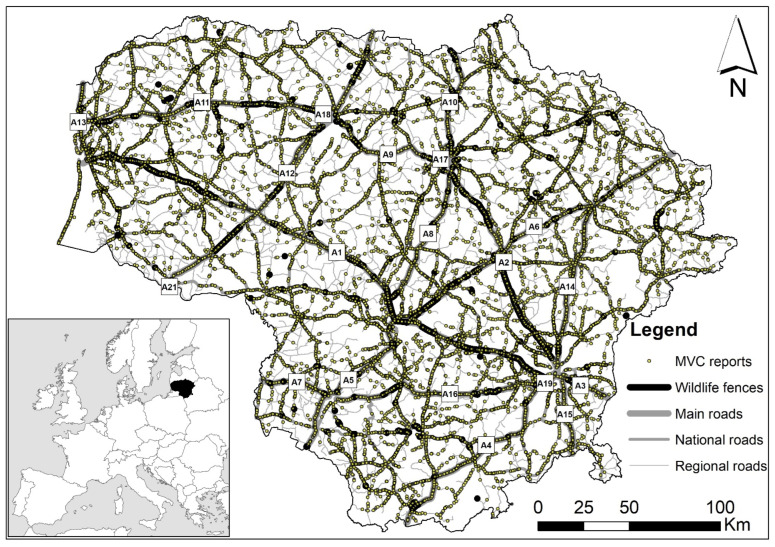
Road network of Lithuania. Main roads are denoted by the numbers A1–A18; road sections with wildlife fences are shown in bold, and locations of mammal-collisions in 2002–2021 are indicated by dots.

**Figure 2 biology-11-00748-f002:**
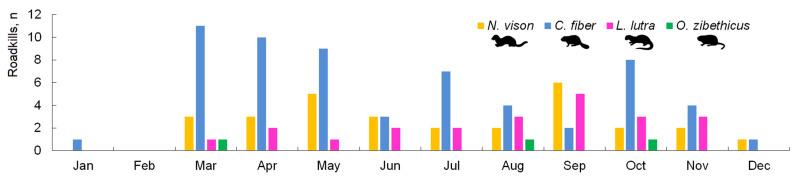
Seasonal distribution of semi-aquatic mammal roadkills, 2002–2021.

**Figure 3 biology-11-00748-f003:**
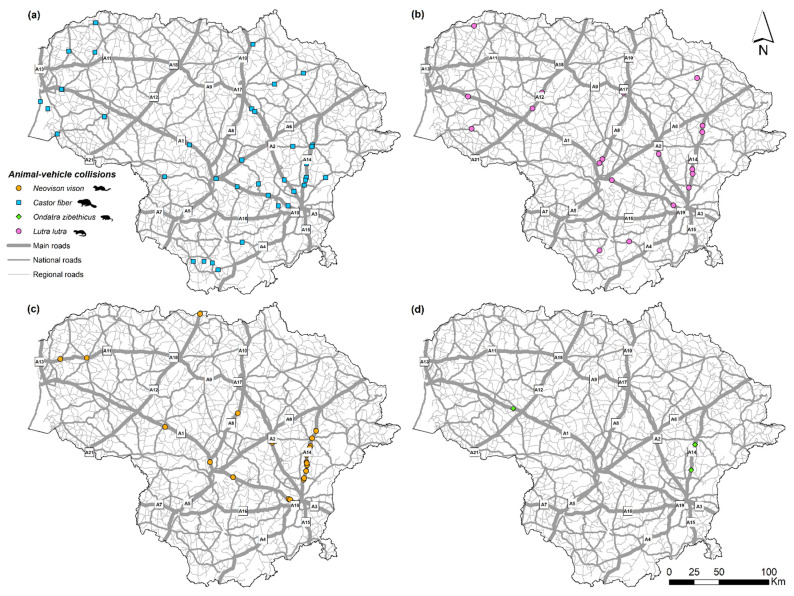
Spatial distribution of the registered semiaquatic mammal roadkill in Lithuania, 2002–2021: (**a**) *C. fiber*, (**b**) *L. lutra*, (**c**) *N. vison*, (**d**) *O. zibethicus*.

**Figure 4 biology-11-00748-f004:**
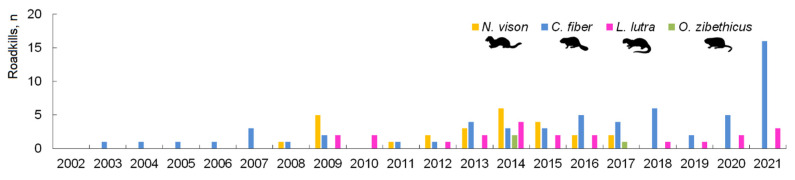
Temporal distribution of the registered semiaquatic mammal roadkill in Lithuania, 2002–2021.

**Figure 5 biology-11-00748-f005:**
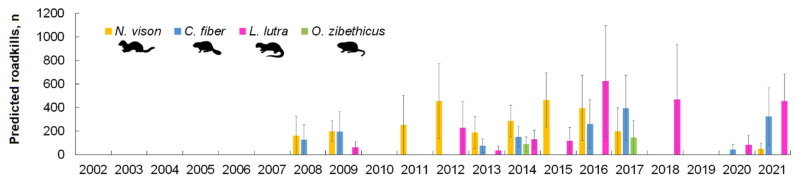
Predictions of unregistered annual semiaquatic mammal roadkill in Lithuania, 2002–2021.

**Figure 6 biology-11-00748-f006:**
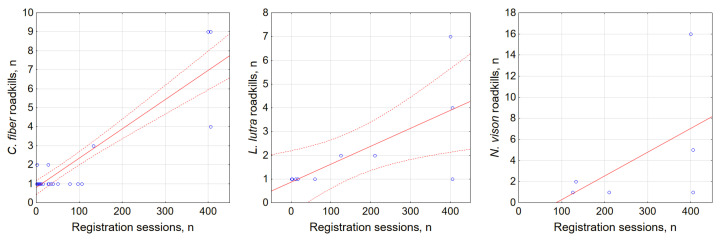
Relationship between registered *C. fiber*, *L. lutra* and *N. vison* roadkill numbers and sampling effort, expressed as the number of registration sessions. Dashed lines show 95% CI, not shown for nonsignificant correlation. Blue dots show registered roadkills.

**Figure 7 biology-11-00748-f007:**
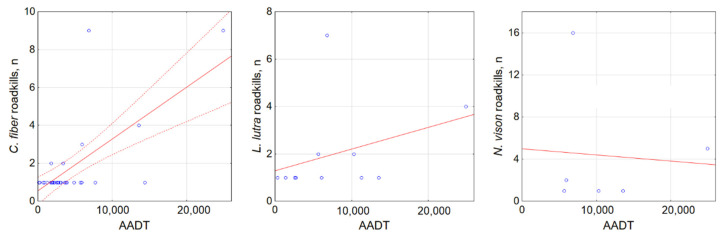
Relationship between registered *C. fiber*, *L. lutra* and *N. vison* roadkill numbers and traffic intensity. Dashed lines show 95% CI, not shown for nonsignificant correlation. Blue dots show registered roadkills.

**Figure 8 biology-11-00748-f008:**
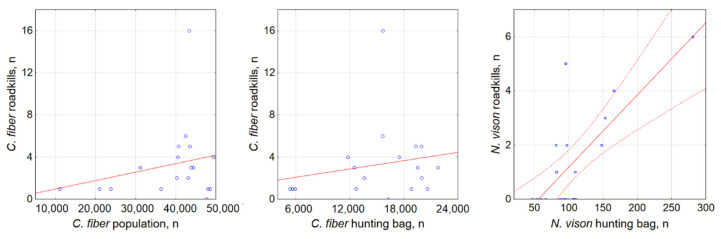
Relationship among registered *C. fiber* and *N. vison* roadkill numbers, population and hunting bag sizes. Dashed lines show 95% CI, not shown for nonsignificant correlation. Blue dots show registered roadkills.

**Figure 9 biology-11-00748-f009:**
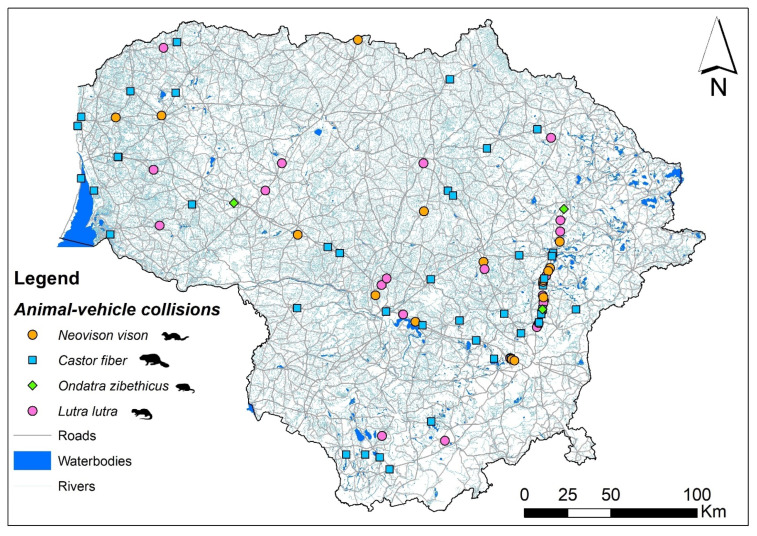
Distribution of the registered semiaquatic mammal roadkill in Lithuania in relation to the water sources.

**Table 1 biology-11-00748-t001:** Population and hunting bag of semiaquatic mammals in Lithuania, 2003–2021 (n/a–data not available).

Year	Beaver (*C. fiber*)		American Mink (*N. vison*)		Muskrat (*O. zibethicus*)	
	Population	Hunting Bag	Population	Hunting Bag	Population	Hunting Bag
2003	11,012	5630	n/a	66	n/a	75
2004	20,926	5890	n/a	92	n/a	31
2005	n/a	4713	n/a	87	n/a	21
2006	23,778	5359	n/a	95	n/a	105
2007	31,160	12,473	n/a	110	n/a	79
2008	36,375	12,702	n/a	109	n/a	60
2009	40,222	13,593	n/a	95	n/a	72
2010	47,702	16,231	n/a	107	n/a	30
2011	48,046	18,821	n/a	82	n/a	72
2012	48,604	20,591	n/a	148	n/a	231
2013	49,446	11,778	n/a	153	n/a	148
2014	44,416	21,749	n/a	281	n/a	329
2015	43,802	19,544	n/a	166	n/a	99
2016	40,618	19,293	n/a	97	n/a	76
2017	40,506	17,503	n/a	81	n/a	73
2018	42,396	15,637	n/a	59	n/a	60
2019	43,148	19,977	n/a	106	n/a	42
2020	43,551	19,907	n/a	46	n/a	39
2021	43,355	15,651	n/a	53	n/a	23

**Table 2 biology-11-00748-t002:** Amount of wild mammal roadkill in Lithuania, 2002–2021, in decreasing order. Semiquatic mammal species shown in bold. Species with roadkill numbers less than 20 are pooled.

Species	N
Roe deer (*Capreolus capreolus*)	25,583
Moose (*Alces alces*)	2457
Wild boar (*Sus scrofa*)	2027
Raccoon dog (*Nyctereutes procyonoides*) ^nn^	1659
Red fox (*Vulpes vulpes*)	1175
Eastern European hedgehog (*Erinaceus roumanicus*)	1158
European hare (*Lepus europaeus*)	739
Red deer (*Cervus elaphus*)	633
Marten (*Martes* sp.)	413
Badger (*Meles meles*)	203
European polecat (*Mustela putorius*)	184
Pine marten (*Martes martes*)	70
**Beaver (*Castor fiber*)**	**60**
Stone marten (*Martes foina*)	35
Red squirrel (*Sciurus vulgaris*)	40
Fallow deer (*Dama dama*) ^nn^	45
**American mink (*Neovison vison*)** ^nn^	**26**
Grey wolf (*Canis lupus*)	25
**Eurasian otter (*Lutra lutra*)**	**22**
European mole (*Talpa europaea*)	20
**Muskrat (*Ondatra zibethicus*)** ^nn^	**3**
Unknown	3313
Other species	46
Total, N	39,936

^nn^ Non-native species; Other species included European bison (*Bison bonasus*), lynx (*Lynx lynx*), raccoon (*Procyon lotor*) ^nn^, least weasel (*Mustela nivalis*), stoat (*Mustela erminea*), mountain hare (*Lepus timidus*), black rat (*Rattus rattus*), Norway rat (*Rattus norvegicus*) ^nn^, bank vole (*Clethrionomys glareolus*), yellow-necked mouse (*Apodemus flavicollis*), common shrew (*Sorex araneus*) and water shrew (*Neomys fodiens*).

**Table 3 biology-11-00748-t003:** Distances among semiaquatic mammal roadkill and the nearest waterbodies or wetlands, presented as average ±SE; min–max (in m). CLC: Corine land cover (1:50,000), GRPK: base cadastre (1:10,000).

	*C. fiber*	*L. lutra*	*N. vison*	*O. zibethicus*
CLC water	2823 ± 371 0–11,310	2563 ± 619 159–10,450	2343 ± 389 338–9704	2376 ± 1085 229–3719
CLC wetland	7067 ± 782 296–21,824	6503 ± 1173 267–20,608	5434 ± 1056 719–27,233	7372 ± 4695 159–16,185
GPRK river	303 ± 101 6–4256	284 ± 63 13–1059	324 ± 64 13–1388	90 ± 10 73–106
GPRK lake	1093 ± 156 13–4116	1260 ± 203 179–2961	1108 ± 193 114–4100	901 ± 547 229–1984

## Data Availability

Due to ongoing investigation, data of this study are available from the corresponding author upon personal request.

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
