# Peer review of "Factors Affecting Roadkills of Semi-Aquatic Mammals"

_biology, 2022, doi:10.3390/biology11050748_

Round 1

Reviewer 1 Report

In the manuscript „Factors Affecting the Diversity of Registered Roadkills of Semi-aquatic Mammals” the authors present and analyze the roadkills and their conditions of four semiaquatic mammal species in Lithuania, based on long-term data sets. The undoubted advantage of the work is the processing of a large amount of data, which covers the whole of Lithuania. The objectives are clearly defined. The overall study is relatively straightforward. However, to be publishable, I suggest the authors apply some minor changes (minor revisions).

In the title, I find the word “Diversity” redundant. Although a list of mammal species affected by roadkills is included, the manuscript focuses on the four semiaquatic species only, no analyses is carried out investigating the effect of factors on diversity.

The “Simple summary “ and the “Abstract” is almost identical. I suggest to simplify and rephrase the “Simple summary” section.

The introduction is well structured. The significance of roadkills is relatively well represented in the literature. The literature cited in this regard is sufficient.

The methods used for data processing and evaluation are appropriate. However, with regard to annual average daily traffic (AADT), it may not be appropriate to use data of a single year (2019) in the regression analysis. The annual average daily traffic cannot be considered constant, this variable also changes over time, and considering the 18-year study period, this change can be significant. It would be worthwhile to find a solution for this problem (correction factors?) if accurate data from each year are not available.

In the Discussion, it would be interesting to read about the Eurasian beaver’s (Castor fiber) history, extinction and subsequent reintroduction, as well as the reasons for the stronger population growth during the last years. Similarly, the rapid population increase of muskrat following its appearance in the country, and the stagnation or decline of its population in recent decades may be interesting adjuncts to understand and explain the results better.

The Conclusion in this form is inappropriate, practically a repetition of the main results. Therefore, it is suggested that this section be rewritten or omitted (according to the journal's guidance, the Conclusion is optional).

Figures

The figures are illustrative and help to understand the results. Some minor suggestions:

  • The two separate diagrams in Figure 2 (Line 130) can be merged. The same can be applied in the case of Figure 4 (Line 206) and Figure 5 (Line 222).
  • Although the colors / symbols indicate the species, I suggest applying the silhouettes used in the previous diagrams (e.g. Fig. 2) also in the four maps of Figure 3, making it easier to identify the species.

Other minor comment:

Line 335 – I think the % value “38–24%” is incorrect, is it maybe “38–54%”? Please, check.

Language:

The English and spelling are basically correct, but I definitely recommend reading and correcting the manuscript by a native speaker. The use of multiple complex sentences is often confusing. There are also some misleading terms in the manuscript, e.g.  Line 35 – “approximation of roadkilled numbers” is more correct as “number of roadkilled individuals”.

Author Response

Rev#1 comments and answers

In the manuscript „Factors Affecting the Diversity of Registered Roadkills of Semi-aquatic Mammals” the authors present and analyze the roadkills and their conditions of four semiaquatic mammal species in Lithuania, based on long-term data sets. The undoubted advantage of the work is the processing of a large amount of data, which covers the whole of Lithuania. The objectives are clearly defined. The overall study is relatively straightforward. However, to be publishable, I suggest the authors apply some minor changes (minor revisions).

Comment: In the title, I find the word “Diversity” redundant. Although a list of mammal species affected by roadkills is included, the manuscript focuses on the four semiaquatic species only, no analyses is carried out investigating the effect of factors on diversity.

Answer: Title changed as suggested

Comment: The “Simple summary “ and the “Abstract” is almost identical. I suggest to simplify and rephrase the “Simple summary” section.

Answer: we simplified Simple summary, now it is different from the Abstract. However, as these two parts of the paper never go together, maybe we can keep some information similar in both sections?

Comment: The introduction is well structured. The significance of roadkills is relatively well represented in the literature. The literature cited in this regard is sufficient.

Answer: thank you

Comment: The methods used for data processing and evaluation are appropriate. However, with regard to annual average daily traffic (AADT), it may not be appropriate to use data of a single year (2019) in the regression analysis. The annual average daily traffic cannot be considered constant, this variable also changes over time, and considering the 18-year study period, this change can be significant. It would be worthwhile to find a solution for this problem (correction factors?) if accurate data from each year are not available.

Answer: mistype, we used average annual daily traffic for every year, and this is clear from the Figure 7. Thank you for the comment, we cleared mistype and changed reference; unfortunately, full information is only on Lithuanian page of the Lithuanian Road Administration under the Ministry of Transport and Communications. In the English version of the mentioned website required information is not presented so far.

Comment: In the Discussion, it would be interesting to read about the Eurasian beaver’s (Castor fiber) history, extinction and subsequent reintroduction, as well as the reasons for the stronger population growth during the last years. Similarly, the rapid population increase of muskrat following its appearance in the country, and the stagnation or decline of its population in recent decades may be interesting adjuncts to understand and explain the results better.

Answer: three small paragraphs were added as per your comment, as well as 2 new references on the muskrat situation. We added short information on the mink, too – though it was not asked.

Comment: The Conclusion in this form is inappropriate, practically a repetition of the main results. Therefore, it is suggested that this section be rewritten or omitted (according to the journal's guidance, the Conclusion is optional).

Answer: conclusions shortened and rewritten, so now is Conclusion only.

Comment: The figures are illustrative and help to understand the results. Some minor suggestions:

  • The two separate diagrams in Figure 2 (Line 130) can be merged. The same can be applied in the case of Figure 4 (Line 206) and Figure 5 (Line 222).
  • Although the colors / symbols indicate the species, I suggest applying the silhouettes used in the previous diagrams (e.g. Fig. 2) also in the four maps of Figure 3, making it easier to identify the species.

Answer: Figures 2, 4 and 5 reworked as suggested, merging all 4 species in one diagram. Colors were also changed, to keep better compatibility with the maps.

We add silhouettes also to legends of the Fig. 3 and Fig. 9. To use silhouettes instead of dots in the map seems not possible, as (i) maps are generated in GIS, where such symbols are not available, (ii) readable figures are obscuring map backgrounds, and (iii) in some roads, silhouettes simply do not fit.

Other minor comment:

Comment: Line 335 – I think the % value “38–24%” is incorrect, is it maybe “38–54%”? Please, check.

Answer: apologies, mistype. Thank you for your thoroughness.

Language: The English and spelling are basically correct, but I definitely recommend reading and correcting the manuscript by a native speaker. The use of multiple complex sentences is often confusing. There are also some misleading terms in the manuscript, e.g.  Line 35 – “approximation of roadkilled numbers” is more correct as “number of roadkilled individuals”.

Answer: we acknowledge this comment, exactly this is “extrapolated number of roadkilled individuals”. By extrapolation we mean predicting of roadkill values outside the range of data we have.

One of the authors, Jos Stratford is a native speaker and language editor. Text was re-checked, some mistypes cleared; we believe it is OK now.

Reviewer 2 Report

Dear editor and authors

The study entitled “Factors Affecting the Diversity of Registered Roadkills of Semi-aquatic Mammals” provides new and valuable data that allow evaluate and discuss roadkill of semi-aquatic mammals in Lithuania, relating these to monitoring effort, traffic intensity, population size and proximity to waterbodies. Data are original, and results are convincing and very relevant for publication. In this sense, manuscript has merits and fits completely within the scope of the journal. I didn't find problems or errors on research design or in methods. The results are clearly presents and discussions and conclusions are supported by results. The overall merit of the manuscript is high. For these reasons, I recommend the accept in present form.

Sincerely,

Reviewer

Author Response

Answer: we have a pleasure to thank you for this great evaluation of our work

Reviewer 3 Report

This is a well-written and interesting manuscript. The analyses are appropriate and the overall length is fine. I assume the Journal editor will follow with the specific format for figures and tables.

Below are some more specific comments.

Simple summary

Line 12 to 13 do you really need this number here?

L 15 What is a hunting bag?

L 45 replace “member of this group” with a better term

L 46 replace including with such as

L 49 and 53 all waterbodies? Maybe use most suitable waterbodies

L 60 what is a bag?

L 93 use long-term

L 101 what are a registration sessions?

L 131 replace “in” with from 2002-2021

179 why is wild needed in comparison to domestic species?

188 Table 2 the species should be organized by family and genus, non-native species could be marked with a (nn) notation. How about domesticated animals?

L 199 Fig 3 It might be nice to indicated water bodies, like in Fig 9

316 replace a close neighbor with a neighboring country

Author Response

Rev#3 comments and answers

Comment: This is a well-written and interesting manuscript. The analyses are appropriate and the overall length is fine. I assume the Journal editor will follow with the specific format for figures and tables.

Answer: thank you for your evaluation. We followed journal Template, manuscript is formatted in line with Journal requirements.

Below are some more specific comments.

Comment: Simple summary, Line 12 to 13 do you really need this number here?

Answer: we deleted mentioned values, this goes in line with comment of Rev#1 – and replaced these with a text “being rare events”

Comment:  L 15 What is a hunting bag? Comment: L 60 what is a bag?

Answer: A bag (hunting bag), in the context of fishing and hunting, is a quantity of fish caught or game killed, normally given as number of animals. Hunting bag statistics are often the only available data for performing ecological studies about harvested species, and total harvest is sometimes used as a proxy of abundance of the game species under study in a given geographical area and period of time.
Apologies, in theriology this is widely used term. So to make this understandable for the other readers, we introduce short explanation at the first use, Line 60, also in the Simple summary and Abstract.

Comment: L 45 replace “member of this group” with a better term

Answer: replaced with “animals”

Comment: L 46 replace including with such as

Answer: replaced as advised

Comment: L 49 and 53 all waterbodies? Maybe use most suitable waterbodies

Answer: In fact, it is as said. Having beaver density in line with carrying capacity, all types of waterbodies (not only most suitable) are inhabited, at least until food resources are exhausted. Even small pools in agricultural landscape may hold a migrating individual or small family, if there are resources available. We cite [9], and this the proper reference for Lithuania.
We changed wording to acknowledge your comment. Now Line 49 reads “In Lithuania, C. fiber is extremely widespread and inhabits nearly all waterbodies, especially those with shores overgrown with deciduous trees and shrubs. It is abundant in land reclamation channels and builds dams to raise water level where required. Even small pools or swamps in the agricultural fields are inhabited, at least temporarily, until there are foods around [9]”.

We also corrected Line 53, it now reads “L. lutra is also widespread and inhabits, at least temporarily, all types of waterbodies with running water, including reclamation channels, as well as lakes and ponds.” Situation is similar – otter is not a rare species in the country.

Comment: L 93 use long-term

Answer: done

Comment: L 101 what are a registration sessions?

Answer: single registration session is any length of the same-numbered road, travelled in the same day. Therefore text is added:
A registration session was defined as registering roadkills along any length of a same-numbered road travelled on the same day. 3570 such sessions were conducted nationally from 2007–2021, the total length of all driven routes equaling 275,300 km.

Comment: L 131 replace “in” with from 2002-2021

Answer: changed with “,”, according to the other captions.

Comment: 179 why is wild needed in comparison to domestic species?

Answer: because we do not analyze roadkills of domestic animals (see next answer).

Comment: 188 Table 2 the species should be organized by family and genus, non-native species could be marked with a (nn) notation. How about domesticated animals?

Answer: Domesticated animals were not included into analyses, as they really are related not to habitat, but to the presence of settlements/farmsteads. However, to give an idea, we present number of roadkills of domestic animals in the text.
Further, your comment is acknowledged by marking non-native species with superscript.
We however would like to keep numbers in the Table 2 arranged descending, to show where are semiaquatic mammals according the number of their roadkills. Hope you will agree with our logic.

Comment: L 199 Fig 3 It might be nice to indicated water bodies, like in Fig 9

Answer: we tried this approach already, however, in such a scale waterbodies become a mess and yields no useful information.

Comment: 316 replace a close neighbor with a neighboring country

Answer: replaced